# User Involvement in the Handover between Mental Health Hospitals and Community Mental Health: A Critical Discourse Analysis

**DOI:** 10.3390/ijerph18073352

**Published:** 2021-03-24

**Authors:** Kim Jørgensen, Tonie Rasmussen, Morten Hansen, Kate Andreasson, Bengt Karlsson

**Affiliations:** 1The Research Collaboration Psychiatric Centre, University of Copenhagen, DK-3400 Hillerød, Denmark; kate.trein.aamund.wuest@regionh.dk; 2Center for Quality and Development, Department of Social Health, Rudersdal Kommune, DK-3460 Birkerød, Denmark; TORA@rudersdal.dk; 3FACT Team 1, Psychiatric Outpatient Clinic Nørrebro Griffenfeldsgade, 46-2200 Copenhagen, Denmark; aesel82@gmail.com; 4Faculty of Health and Social Sciences, Department of Health, Social and Welfare Studies, Center for Mental Health and Substance Abuse, University of South-Eastern Norway, 3679 Notodden, Norway; Bengt.Karlsson@usn.no

**Keywords:** mental health hospitals, community mental healthcare, discourse analysis, handover, user involvement, governmentality

## Abstract

Introduction: This study aimed to explore how healthcare professionals and users could perceive user involvement in the handover between mental health hospitals and community mental healthcare, drawing on the discourse analysis framework from Fairclough. Methods: A qualitative research design with purposive sampling was adopted. Five audio-recorded focus group interviews with nurses, users and other health professionals were explored using Fairclough’s discourse analysis framework. Ethical approval: The study was designed following the ethical principles of the Helsinki Declaration and Danish Law. Each study participant in the two intersectoral sectors gave their informed consent after verbal and written information was provided. Results: This study has shown how users can be subject to paternalistic control despite the official aims that user involvement should be an integral part of the care and treatment offered. As evidenced in discussions by both health professionals and the users themselves, the users were involved in plans with the handover on conditions determined by the health professionals who were predominantly focused on treating diseases and enabling the users to live a life independent of professional help. Conclusions: Our results can contribute to dealing with the challenges of incorporating user involvement as an ideology in the handover between mental health hospitals and community mental health. There is a need to start forming a common language across sectors and, jointly, for professionals and users to draw up plans for intersectoral care.

## 1. Introduction

In several Western countries and regions such as the USA, Australia, the United Kingdom and Scandinavia, user involvement has been a milestone in the health sector. Health professionals must involve users in healthcare across mental health hospitals and community mental healthcare [1,2]. The Health Act in Denmark emphasizes that the patient’s right to self-determination must be respected [3,4]. Therefore, the patient must have information about their illness and treatment in order to be able to make choices and make commitments before initiating treatment [5]. In a study, involvement is equated with the dissemination of knowledge to the patient—for example, [6]—but what role should the patient play in addition to receiving knowledge and giving their consent to the treatment? Involvement is a political goal [5], where the patient must play the main role in intersectoral care. However, it is not clear what this role entails [7].

Therefore, it must be up to the user and the healthcare professional to reconcile the expectations for involvement. This process may sound simple, but in practice, the user’s encounter with the healthcare system can only be described to a small extent as an encounter between two people. Therefore, the question of what involvement entails extends beyond the specific meeting between a user and a healthcare professional, as well as locally among managers and healthcare professionals, where there must be a framework for involving users individually in their treatment process. Otherwise, users will be exposed to involvement at the discretion of healthcare professionals, which is not necessarily in harmony with their wishes [8,9].

Healthcare professionals must share the responsibility for treatment with the patient and not deprive them of the responsibility but facilitate the patient’s resources through the treatment process. This situation means that healthcare professionals must focus on the patient’s resources and try to fill in and qualify where the patient does not have the necessary capacity for self-care. Let us suppose that this vision is to be implemented in the healthcare system. In this case, however, it will involve a reform of the healthcare system, where organizational structures are arranged differently to adapt to the patients’ wishes [10,11].

The patient must also have a real opportunity to make individual choices if they were to take joint responsibility [12]. Healthcare professionals have the obligations of contributing knowledge and competencies that enable a user to take responsibility for their life situation and actively contribute to the treatment [13,14]. The introduction of user involvement is also a result of the desire of different user organizations for the user and relatives to become a more active part of the treatment, for individual needs to be listened to and for a treatment course to be planned together [15].

The discourse of involvement has become a symbol for how patients should take greater responsibility for their health and treatment, e.g., to relieve the healthcare system in the future. The more a patient can take responsibility for their situation and health, the less they will use health services.

The development of the health service has moved in the direction of an individualistic approach. The responsibility for recovering from illness is largely placed with the patient who is expected to familiarize themselves with their illness and treatment and make an active effort to recover [4]. According to the Danish Health Act, health professionals must ensure that the patient obtains an informed knowledge base in order to be able to decide whether they will give informed consent to a given treatment offer [16,17].

There is no authorized definition of user involvement. However, it can be seen as an attempt to reduce social inequality, promote patients’ self-care and the struggles of semi-professionals for influence or as an attempt to reduce costs in the public system by assigning patients with greater responsibility for treatment [6,18]. User involvement can be considered a democratic development in the healthcare system, where the user has an equal voice in decisions about their treatment. The health system offers the user the chance to be involved in the organization of care. However, another interpretation is that the user must, to a greater extent, consider themselves to be at the center in the sense of taking responsibility for their health and life treatment situation and demand the necessary help to regain their self-care. Self-care depends on the individual’s resources, abilities and conscious desires. Self-care is emphasized as the individual’s responsibility [19,20,21].

As described, different meanings are applied to user involvement, and each meaning can be understood as different discourses. Different linguistic expressions involving different notions of involvement and specific user options for action are positioned in interprofessional social practice, and notions are based on being decisive for how practice is carried out. Therefore, the ways a user is specifically involved in care and treatment differ. Few articles show the discourses articulated in the handover between a mental health hospital and community mental healthcare setting. Drawing on the discourse analysis framework of Fairclough, this study aimed to explore how healthcare professionals and users could perceive user involvement in the handover between mental health hospitals and community mental healthcare.

## 2. Methods

The study enters a social constructivist framework in which inclusion is not seen as an objective quantity in the sense of being immutable and universal; involvement, on the other hand, is only accessible to us through the social interaction process and it is through this that we experience and recognize it [22,23]. Our knowledge does not reflect the world as it is in itself. It describes the world in terms of the social and cultural contexts that enable us to participate, influencing our understanding of the world. From a social constructivist perspective, there is a strong argument for taking a critical approach to conventional and obvious ways of understanding and acting in the social health field [24,25].

The discourse on user involvement can be seen as a multidimensional and social practice which unfolds under specific political, socio-cultural, economic and historical circumstances. Accordingly, this study is designed as a critical discourse analysis and aims to shed light on the linguistic-discursive dimension of social and cultural phenomena, as well as on the processes of change in late modernity. This will show how discourses are practiced to sustain the social world and social relations, which inevitably involve uneven power relations [26,27]. A discourse is understood as a certain way of talking about and understanding the world or a part of it, where language is seen as a form of social practice rather than an individual activity. Discourses constitute both the social world and other social practices; discourse is in a dialectic relationship with other social dimensions; discourses add to, form, transform and reflect social structures and processes [26,27].

### 2.1. Recruitment and Sampling

We used a purposive sampling technique to include participants [28] and we obtained permission from the management in both sectors to invite participants to the project [29]. Healthcare professionals had to have experience and/or knowledge of treatment and care for users with mental disorders. Users were required to have experience with admissions at least in terms of a mental health hospital or community mental healthcare.

In total, 27 informants agreed to participate, nine of whom represented a user perspective. Among the healthcare professionals, 14 assisted with treatment in inpatient or outpatient psychiatric centers and four were employed in community mental healthcare settings (Table 1).

### 2.2. Data Collection: Focus Group Interviews

Data were collected through open interviews in each of the five focus groups to uncover the informants’ perspectives in depth [30,31]. Focus group interviews sought to achieve a dynamic dialogue and insight into nuanced understandings on the matter. We used an interview guide with sensitive taboo topics to make it easier to express views that are usually withheld (Table 2). Five focus group interviews with four to seven participants were planned—two with health professionals from a mental health hospital, one with health professionals from a community mental healthcare setting and two with users. The focus group interviews lasted about an hour and were conducted in the environment preferred by the participants.

The first author held all interviews with the second or third co-author. The first author was the moderator and asked questions while the other assisted with supplementary questions and validating the content at the end of the discussions.

The interviews were transcribed by a research assistant and resulted in 71 pages of text for the analysis. All authors were given the opportunity to review and approve these verbatim transcripts. The research group reflected on and discussed our positions in the research field.

As a research group, we have longstanding clinical experience and/or knowledge about the handover between mental health hospitals and community mental healthcare settings. Three authors are nurses, one a physician and one an educator with user background. Three have PhDs, a postdoc, a senior researcher position, a professor position and one of the nurses has a master’s degree.

### 2.3. Data Analysis

Fairclough applies the concept of discourse in three diverse ways: (1) Discourse refers to language use as social practice; (2) discourse is understood as the kind of language used within a specific context and (3) discourse refers to a way of speaking which gives a sense of meaning from a particular perspective. He states that discourse contributes to constructing social identities, social relations and knowledge and meaning systems from this background. Accordingly, discourse has three functions: an identity function, a relational function and an ideational function. Fairclough sees every instance of language use as a communicative event consisting of three dimensions: (1) a spoken or written language text, (2) a discursive practice which involves the production and interpretation of the text and (3) a social practice [4,18,25]. All three dimensions should be covered in the analysis of a communicative event [22,26,32]. In Fairclough’s understanding, texts may never be understood or analyzed in isolation. They can only be understood in connection with the meaning in other texts and their social context; discourses are dialectic [27,32]. Consequently, we analyzed the texts using a three-dimensional framework: as a text, as a discursive practice and as a social practice.

(1) The text analysis. At this level of analysis, the analyst should consider various linguistic aspects of the text. Accordingly, we read each text several times to grasp how user involvement in the handover between mental health hospitals and community mental healthcare settings is textually specified. We analyzed the texts line by line and word by word. Firstly, we focused on the vocabulary; words and wording describe user involvement. These texts were entered into In Vivo, a computer program showing the textual syntactic horizontally and each text vertically so meanings could be constructed. Secondly, we focused on the grammar of the texts. We also searched for interactional control, transitivity and modality of the texts inspired by Fairclough’s main analytical questions [4].

(2) The discourse practice. Inspired by Fairclough, we focused on the texts’ production by analyzing “interdiscursivity” and “manifest intertextuality”. Concerning text distribution, we analyzed “intertextual chains” and for text consumption by analyzing “coherence” [4,27].

We read the texts to specify which discourse types should be drawn upon and how this should take place. We wanted to specify what and how other texts are drawn upon in the sense of the texts being analyzed.

In addition, we analyzed the texts to show how they include intertextual chains and how the interpretive implications of the intertextual and interdiscursive properties of the texts manifest themselves.

(3) The social practice. The general aim at this level of the analysis was to specify the nature of social practice, part of which is the discourse practice. This explains why the discourse practice is as it is [4].

Accordingly, we discussed the discourse practice to interpret the social and hegemonic relations and structures, the orders of the discourses and the ideological and political effects of the discourses in the main interpretation of the social practice of user involvement in mental healthcare (Table 3).

Ethics

This study was conducted to adhere to the ethics of scientific work. The study was approved by The Danish Knowledge Centre study for Data Reviews [J.nr.: P-2019-753]. According to the Helsinki Declaration [33] and Danish Law [34], no formal permit from a biomedical ethics committee was required. The purpose of the research was not to influence the informants either physically or psychologically. Informed consent was obtained from all participants after receiving verbal and written information about the purpose of the study. The participants were informed that participation could be stopped at any time and that all data would be treated to prevent access to the material by unauthorized persons. Data were treated confidentially and anonymously and audio files were erased after the transcription of the material. All other material containing data was to be destroyed after publication.

Findings

We will present the findings from the three-dimensional analysis following Fairclough’s framework of analysis. The text will occasionally illustrate the development of the discourses found, reflecting the participants’ perceptions of user involvement in the handover between mental health hospitals and community mental healthcare settings.

The vocabulary of the texts according to the participants’ perceptions of user involvement in the handover between mental health hospital and community mental healthcare

Professionals from both sectors used language that discusses an organizational structure such as “outpatient clinics”, “emergency team”, “open ward, “closed ward”, “treatment conference” and “network meetings”. Their language points out structural aims such as “self-support”, “achieving a higher functional level” and “accessing education and a job”. Different methods were articulated as “problem aim lists”, “safety plans” and “cognitive schemas”. Furthermore, they talked about “paradigm shifts”, “recovery”, “holistic assessments” and “peer support”.

The user’s vocabulary was oriented to mental health hospitals with words such as “depot medicine”, “patient-controlled bed”, “regular bed”, “paranoid schizophrenia”, “psychiatrist”, “psychologist”, “social worker”, “home-supporter” and “be approved for support/help”, “paragraphs”, “job consultant” and “abuse of power”.

There are several common features in the language of the professionals and the user language in addressing an institutional language with professional groups, types of psychiatric treatment wards, medicine and professional trading tools.

The wording around user involvement in the handover between mental health hospitals and community mental healthcare was steered by the healthcare professionals’ perspectives as the following quotation shows:


*“They are involved in the treatment here. Then, for me, it’s about talking about help for self-help and aims for the future. Constantly talking about what should happen when you’re not here? What do you want from your life? That’s how I think about being involved” (Interview number (IW 1))*


The context of “involving” was articulated as an organizational aim for what users should be able to do during hospitalization. The user talked about the asymmetric relationship and puts involvement in perspective with a lack of resources when the disease has worsened.


*“Then, it simply came to our notice. When you feel bad, it can be really hard to find resources. Then, you can stand in a slightly unequal position in relation to this. Under normal circumstances, a person might have been able to find more energy. But because the disease may take over, it is really difficult” (IW 4).*



*“For example, if you are hospitalised, it can be very different whether they involve you in what they do at all. I have been in this situation many times: well, what should I do now? Well, you just have to do nothing. You just have to take care of yourself. So, I feel a little divided when I’m hospitalised because I want to know what’s going on. Of course, I join those who meet, but I do not always feel that I am told what is going to happen. They can say: in four weeks, there may be something else, but what then in the next four weeks” (IW 4).*


The interpretive perspective and keywords were framed in a professional manner, where the user becomes a means of achieving the professionals’ aims. The users do not feel that their need for involvement is coordinated. There may be a mismatch between the professionals’ wishes and the users’ expectations of getting involved.

The grammar of the texts: Interactional control.

Interactional control frames the relationship between speakers, e.g., those who set the conversational agenda [27], and is a point of contention between professionals in the mental health hospital and community mental healthcare. Among the healthcare professionals, interactional control was articulated as the right to decide over the user’s plans as the following quotations show:


*“In a mental health hospital, we often think we know what is best for the user’s future. We think, for example. users must be discharged and offered a housing community mental healthcare or that they must be offered support through a social worker. The municipalities dislike it when we come up with all these proposals. It is legitimate to say that the patient is ill and has a low level of function, has difficulty coping with personal hygiene and difficulty with self-care and no network, and then the case that the caseworker would like to say “it sounds as if that you need housing allowance”. In the same ways, the municipality cannot say “Why do you not give that patient some more medicine. A physician would think it was quite provocative” (IW 1).*


Interactional control frames the relationship between the professionals in both sectors. The professionals are preoccupied with the language they use to achieve what they find essential for the users. The language follows unspoken logical structures for what must be implemented for the other party to find solutions for the users themselves. Rather than involving the users by giving them a voice, the focus is on following specific power structures and avoiding creating conflicts with cross-sectoral partners.

The grammar of the texts: Transitivity

When analyzing transitivity, the focus is on how events and processes are connected with the subject and objects [26]. The professionals from both sectors used a kind of transitivity where they spoke positively about user involvement. There is a long way to go from this attitude to see how the individual user is involved in the plans that are made for their future. The professionals pointed out disagreements about how psychiatric services should be organized, and this comes, among other things expressed, as the following quotations show:


*“When we have a long-term period in outpatient clinics with a user, the patient becomes admitted to a mental health ward and all of a sudden there are a thousand people who think that a whole lot needs to happen. Suddenly, there are a thousand professionals who have an opinion on a new plan without necessarily involving outpatient clinics. Also, we can try to make a new plan here where we invite collaborators but they do not want to come along” (IW 1).*


Organizational structures and difficult intersectoral communication are articulated as steering elements for who should be involved in planning users’ intersectoral care, for example.


*“In the hospital sector, our work is interdisciplinary and we try to have a structure. But when you then move into a municipal picture, the job centre only has a certain size. Those who grant housing offer a different dimension. If you also need support, then it is the third case. It’s almost impossible to move in this. If we have to communicate, then it is almost impossible to get hold of the municipalities because they limit phone times down to virtually no availability” (IW 2).*


The involvement of a user’s voice is not highlighted, but rather the professionals focus on the challenges of establishing intersectoral collaboration on the users’ plans. In this connection, the users are mentioned as a means for the professionals to achieve different goals.

The professionals expressed that they involve the users’ perspectives, but there is no common understanding among professionals about what user involvement entails. Users do not feel that they are sufficiently involved in the plans for their care as the following quotations show:


*“We can say that we are a vulnerable group. It can be really hard when you feel bad about finding resources and are not involved. Then, you can stand in a slightly unequal position in relation to the professionals. However, because the disease may take over, it is really difficult” (IW 4).*



*“If you are hospitalised, it is very difficult to be involved. I have been here many times: well, but what should I do now? Actually, you just have to do nothing. You just have to take care of yourself. So, I feel a little split when I’m hospitalised because I want to know what is going on” (IW 5).*


The users perceived the healthcare professionals in a paternalistic form of steering, where users became passive subjects and healthcare professionals decided on their treatment and rehabilitation without their involvement.

The grammar of the texts: Modality

Analyses of modality focus on the speaker’s degree of affinity or affiliation with their statement [26]. The professionals use an objective modality when they talk about user involvement in the handover between the mental health hospital and community mental healthcare. Professionals have a great affinity for treatment plans, patient plans and safety plans, for example, that are articulated using tools. The professionals focus on obligations such as symptom treatment, allocation of aids in community mental healthcare and legislation as the following quotations show:


*“I do not talk about a diagnosis with patients. I talk about symptoms. When you have certain symptoms, you have difficulty with various things. I just think that the community mental healthcare setting’s approaches are different, but it is the same thing we are working around” (IW 3).*



*“When I had to make a paragraph 104 application to an activity centre for the user, the social worker said as a final remark: there is one thing you must know and this is you should always good to be good friends with your caseworker because it is us who hold the purse strings” (IW 4).*


User involvement is subject to some structural power relations that become the framework for how users are involved in the handover between the two sectors. The professionals use some truth modality with high affinity such as:


*“If you have a diagnosis of schizophrenia, it can be very difficult to learn something new. But what you then learned in childhood and adolescence sits there. Having to learn something new can be difficult. Something is happening on the cognitive function level” (IW 5).*


Other examples of truth modality with high affinity are:


*“It is a big problem that caseworkers lack the knowledge and they have no understanding of what challenges the citizens have or what they are dealing with. They simply do not understand it” (IW 3), “We involve patients everywhere.” (IW 1), “We do not hold meetings without the patients’ participation” (IW 1), “they are involved in the treatment here. It’s about talking about help for self-help and goals for the future” (IW 1).*


The users applied a critical language when they talked about user involvement, such as:


*“It has been really difficult that the community mental healthcare and the mental health hospital do not talk to each other. We have to constantly explain our story to everyone” (IW 5);*



*“Then, you have to start over where I have been sitting and thinking they cannot just read the journal to find out about it all” (IW 5);*



*“I have experienced many shifts where social workers do not get involved and they have not talked together about my case” (IW 5).*


Discourse practice

The linguistic analysis identified several interrelated dominant discursive patterns and represented interdiscursivity and intertextuality in the overall coherent interpretation across the focus group interviews. User involvement in the handover between mental health hospitals and community mental healthcare settings drew on manifest intertextuality and intertextual chains by drawing on another communicative event.

The intertextual analysis led to a paternalistic discourse. The professionals’ use of analytical management tools such as treatment plans, patient plans, safety plans and various legislation sets the framework for the users’ opportunities for involvement. The professionals express a self-understanding that they already involve users in the handover. The paternalistic discourse is expressed when the professionals become preoccupied with using their own methods. For example, cognitive techniques are used to promote the patient’s self-care, and by applying cognitive behavioral therapy, the user perspective is automatically involved. As focus groups show, the users do not perceive that they were involved enough and not all the methods made sense for them. The asymmetric power relationship between the professionals and users is clarified. Users become dependent on the methods and theories the professionals choose as a treatment approach and their goodwill to involve them. The lack of clarification on what user involvement entails has the consequence that it is up to individual professionals to make decisions when it comes to involvement.

A common challenge with cross-sectoral collaboration was communication about the users’ plans—for example, in connection with discharge to community mental healthcare. A communitive anxiety discourse assumed that the professionals from both sectors should make sure that they use the correct language when they meet with partners from mental health hospitals or community mental healthcare. The communitive anxiety discourse reflected a tense power relationship between the professionals in the two sectors and the fear of bidding, with concrete proposals that may be perceived as a disrespectful disregard of the other sector’s professionalism, education and power. The professionals advocate for the users’ situations and challenges and told stories about their situations so that partners from the other sector themselves could arrive at possible solutions. The professionals legitimize language codes within each sector and language use does not include the individual users’ voice. In some cases, this can result in less meaningful collaboration between the sectors. The language use can also result in the individual users’ voices not being included.

Even though the professionals distance themselves from a dualistic view of the human being, many examples show how the biomedical discourse dominates the professionals’ focus on users in mental health hospital treatment. The patients’ diagnosis and symptoms are predominantly the subjects of efforts made and medication is attributed to a great deal of power to reduce difficult mental symptoms. Similarly, there is a legislative discourse used by the professionals in community mental healthcare. This is a discourse where users’ mental and social difficulties are interpreted into legislative sections governing what offers users can obtain. This discourse reflects how users’ problems are seen concerning limited financial frameworks for the funds that are provided for impaired physical or mental functioning or particular social problems. In this regard, the professionals talked about the importance of using a language that can translate users’ problems into the language expected by the legislation so they can be granted support and help for the user.

Social practice

In social practice, the three-dimensional analysis maps the social and cultural relations and structures that constitute the wider context of the discursive practice. Our focus turns to the broader social practice that these dimensions belong to [23].

The discursive practice shows the dominance of organizational structure and governing discourses that can be considered from government theory [35]. Discourses that aim to empower citizens to become entrepreneurs and achieve self-governance in their own lives are promoted through empowerment programs. The professionals’ steering tools are legislation and a strong biomedical tradition to solve illness-related problems and make the citizens self-sufficient people who can cope with their own problems without professional help. A cross-sectoral powerful space unfolds between the professionals, where negotiations about a patient’s future occur and where there is no definitive truth as to whether the biomedical and evidence-based approach takes precedence or is weaker as an argument against a social psychological approach. This power game puts the user in a subordinate position where their fate lies in the hands of the professionals. User involvement is drawn on a humanistic understanding but is constituted as a rational and objective approach to understanding and solving problems. Social practice shows an example of the governmentality that unfolds as to how knowledge and power are connected with certain consequences in modern society [35]. Users are considered passive recipients of help. This consideration can be related to the wider social practice that Fairclough terms the “marketisation of discourse”—a social development of late modernity where market discourse colonizes the discursive practices of public institutions [26]. We will further discuss the relation to power in social practice and user involvement in the handover in intersectoral collaboration.

## 3. Discussion

Despite the professionals’ desire to involve users, the users did not experience becoming sufficiently involved. Paternalistic and rational forms of governmentality dominated and overshadowed the possibility of involving users in the best ways possible. The users desired to be more involved in decisions regarding plans with the handover between mental health, hospitals and community mental healthcare settings. Professionals use rational management to promote the users’ ability to be autonomous individuals responsible for their own situation. Cognitive methods are used in mental health hospitals to imply a rational knowledge base that enables the professionals to calculate and manage the users’ offers for care and treatment in order to become an autonomous individual. This kind of appropriate regulation of economy and efficiency is a hallmark of the state management mentality. Governmentality reflects this practice where language is associated with technical tools, practices and institutions aimed at shaping and reshaping behavior [35]. As other studies show, the user is, however, not sufficiently involved [1,6,18] and finds that the professional paternalistic discourse deprives them of the right to influence [4,18,25]. Users’ expert knowledge is downplayed and not included in the rational language of professionals, which elevates medical knowledge, evidence and selected legislative paragraphs as steering tools.

In this study and others [21,36], professionals perceive their work as belonging to a humanistic tradition where users’ perspectives are central. Users’ wishes to live a satisfying life despite any mental or social difficulties form the starting point for the planning of the handover. In the Western welfare health system, the user must be seen as a partner and involved in order for the treatment to reach a higher goal. However, the asymmetrical relationship is also an inevitable factor that will always exist due to professionalism and knowledge [37,38]. However, the professionals’ language reflects that they use a paternalistic discourse where a biomedical discourse gains ground as prevailing in how user difficulties are to be interpreted and resolved. The paternalistic and biomedical discourses offer rational management tools and a strong medical knowledge base formalized on evidence. Discourses that rank high in the hierarchy meet the language used within a legislative discourse, where legal paragraphs use objective criteria for awarding remedial measures in a municipal context.

Mental health hospitals predominantly employ professionals with medical backgrounds. This fact is reflected in their language and their perspectives when they contribute to communication with colleagues and users connected with the handover.

Similarly, community mental health professionals represent a composition of professionals mainly with a social–psychological background, for example, social workers and educators, with fewer nurses. Cross-sectoral collaboration occurs between medical and social–psychological knowledge traditions, during which a communitive anxiety discourse emerges. There seems to be a predominant linguistic caution where professionals from each sector try not to show disrespect. Each sector has a self-legitimate patent on language and the right to decide the plan they think is best for users. This linguistic consideration is about delivering a narrative to the professionals from the other sector. The professionals in the collaborative sector themselves developed the idea that the other sector had intended a solution, but through fear of showing disrespect, they could not present this.

In light of user involvement, the user becomes the key and is responsible for ensuring coherence across sectors. The user is subject to mental healthcare drawn on rational and effective values and must live with the terms of service delivery despite it not necessarily matching individual needs. As mentioned, the term governmentality, taken from Foucault, is often used to understand power relations as a constructive process [18]. Users with mental illness are often weak and vulnerable. In the Western welfare health system, discourses dominate such as, for example, efficiency, self-responsibility, freedom, empowerment and self-help [39]. The user has become a management tool in the healthcare system. The aim is to have active, competent and responsible users and a healthcare system that meets the users’ demands and expectations. This development is not primarily linked to humanistic ideals but to efforts to provide quality and create efficiency in the healthcare system, i.e., neoliberal governmentality ideals [18]. User involvement is constructed as a term that the professionals consider as achieved across the professional environments and methods. Users experience being treated as objects and that they have to adapt to the system’s requirements to obtain help, even if much does not make sense. They do not experience it as legitimate to be ill where care becomes an aim in itself. Instead, they experience being pushed into job training and pension cases dragged out for a long time while trying to survive with difficult mental health symptoms.

### 3.1. Methodological Considerations

Fairclough’s critical discourse analysis has provided a systematic tool that has made it possible to generate discourses and view these in a larger social practice. To increase the credibility of the discourse analysis, we have supported the study’s trustworthiness and transferability by including quotations as examples of the analysis.

To ensure this study’s validity, we carefully selected rigorous data material and systematically treated the material in depth. The results are in line with existing research. They are transferable to clinical practice. They contribute to a greater understanding of the challenges associated with healthcare professionals and users’ perceptions of user involvement in intersectoral collaborations between mental health hospitals and community mental health.

### 3.2. Limitations

The data consisted of five focus group interviews, and in some of these, we did not manage to obtain our desired number of ten participants. We chose to conduct two focus groups with seven attendees, one with five and two with four. The groups with four and five participants did not achieve the same dynamic conversation that typically characterizes focus group interviews. However, despite the few informants in some of the focus groups, these smaller groups ended up providing rich answers that helped form the basis of the analysis. To ensure the validity of this study, we carefully selected rigorous data material and systematically treated the material through in-depth exploration using critical discourse analysis.

## 4. Conclusions

This study has shown how users can be subject to paternalistic control despite official aims; user involvement should be an integral part of the care and treatment offered. As evidenced in discussions by both health professionals and the users themselves, the users were involved in plans with the handover on conditions determined by the health professionals, who were predominantly focused on treating diseases and enabling the users to live a life independently of professional help.

Health professionals from mental health hospitals and community mental healthcare settings always wish users have the best user involvement. However, they have no common understanding of user involvement. This study reveals that health professionals believe that they involve users, but several challenges appear when trying to achieve this aim. There is a need to start forming a common language across sectors and, jointly, professionals and users, as well as to draw up plans for intersectoral care.

## Figures and Tables

**Table 1 ijerph-18-03352-t001:** Overview of the focus groups.

Focus Group Number	Context	Number of Participants Per Focus Group
1	Mental health hospital	Healthcare professionals *n* = 7
2	Mental health hospital	Healthcare professionals *n* = 7
3	Community mental healthcare	Healthcare professionals *n* = 4
4	Mental health hospital	Users *n* = 4
5	Community mental healthcare	Users *n* = 5
	total number of participants: 27	

**Table 2 ijerph-18-03352-t002:** Taboo topics.

General perceptions and experiences of intersectoral care in mental health
Examples of intersectoral care
User participation
Trust
Communication
Relatives
Continuity and coherence
Short resumé
Examples of in-context experiences
Structure
Continuity
Transitions
Individually focused
Meanings/confidences/relationships
Final summary

**Table 3 ijerph-18-03352-t003:** Data analysis.

Text Analysis
Vocabulary: Vocabulary use of words, keywords
Grammar
Interactional control: How is control negotiated?
Transitivity: Are passive clauses or terms of nominalization frequent, and if so, what functions do they serve?
Modality: What sort of modalities are most frequent, subjective or objective?
Discourse practice Text production
Interdiscursivity: What discourse types are drawn upon in the texts, and how?
Manifest intertextuality: What other texts are drawn upon in the constitution of the texts, and how?
Text distribution Intertextual chains: What sort of transformation does this (type of) discourse sample undergo—is it stable, shifting or contested?
Text consumption: Coherence: What are the interpretative implications of the intertextual and interdiscursive properties of the texts?
Social practice
What is the nature of the social practice that the discourse practice belongs to—why is the discourse practice the way it is?

## Data Availability

No new data were created or analyzed in this study. Data sharing is not applicable to this article.

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
