# Peer review of "User Involvement in the Handover between Mental Health Hospitals and Community Mental Health: A Critical Discourse Analysis"

_ijerph, 2021, doi:10.3390/ijerph18073352_

Round 1

Reviewer 1 Report

The presented manuscript is very interesting and highly relevant. It is worth paying attention to user participation in the process of adapting hospitals to the community context. 

The paper is also very well written. And its positioning can be extremely useful to activate other types of mental health practices. 

Below, we indicate a series of recommendations to improve the manuscript presented:
- It would be interesting to include bibliographical references that can sustain the approaches of the first sentences of the methodology. 
- It would be very useful for the authors to reflect on the limitations of the study and the possibilities for the future based on the work presented.

Author Response

Dear reviewer

We thank you very much for your great effort to read our paper and for your constructive comments that we have tried to accommodate in our next draft.

We have also sent it to prof editing to strengthen the language.

Best regards 

Kim Jørgensen

Tonie Rasmussen

Morten Hansen

Kate Andreasson

Bengt Karlsson

Reviewer 2 Report

Here, Kim et al. conducted a study that they investigated how the mental health professionals and the users thought about the patient involvement during the treatment, especially when they (the patients) were transferred between the mental health hospital and community mental healthcare. The author did interviews with five focus groups and conducted the research following the ethical principles. The conclusion of this study is the health professionals play a dominant role with a focus on illness treatments and there is a lack of patients’ involvement during the handover process. The manuscript is overall well written while some sentences are too long to read. It will be very helpful for the readers if the authors can revise some sentences. Otherwise, I think the manuscript is ready to accept.

Author Response

(The authors gave the same response as above.)

Reviewer 3 Report

The authors use focus groups to shed led on the user's and professional's perspectives on user involvement in mental health care, specifically the transition from hospitals to community mental health care. Results are shown in short excerpts and suggest that type and extent user involvement is largely dictated by the professionals in a 'paternalistic' fashion and sometimes even neglects the individual's right and duty for self-determination, despite it being deemed so important in the Danish Health Act. The manuscript addresses this important issue with the sophisticated method of focus groups and thus should be published. However, the data analysis method and the presentation of results needs improvement so that readers do not get lost in the manuscript. At least, direct citations should be italicised oder put in between quotation marks. Above and beyond mere readibility, I strongly suggest the authors find a way to summarise the most important points in figures (e.g., the results) and tables (e.g., the Fairclough distinctions), to make the wealth of information more processable for the reader. 

Other than that, the manuscript would profit from a thorough proof-reading which does not only include typos and wording mistakes but also checks for ease of understanding / understandability. I applaud the authors on so proficiently maneouvering complex theoretical notions, however, I think they could make more of an effort to reduce unnecessary complexities, e.g. in sentence structure and length. First and foremost, that would include getting rid of mistakes like typos and such, e.g. in line 182 / 183, „One works as an educator in a mental health centre and has experience as a service user—three with PhDs and the other with a master’s degree“, or lines 390f., user involvement (…) „drew on manifest intertextuality intertextual chains“. 

Note that while the suggested changes might seem a lot, I do not see that the manuscript needs changes in content - its message seems worthy of publishing as is. All I would like to suggest is to increase the readability, so that a wider audience, including practitioners in particular, can profit from the research. 

Author Response

(The authors gave the same response as above.)
